# Memory and decision making interact to shape the value of unchosen options

Natalie Biderman [1✉] & Daphna Shohamy[1,2✉]

The goal of deliberation is to separate between options so that we can commit to one and leave the other behind. However, deliberation can, paradoxically, also form an association in memory between the chosen and unchosen options. Here, we consider this possibility and examine its consequences for how outcomes affect not only the value of the options we chose, but also, by association, the value of options we did not choose. In five experiments (total $n = 612$), including a preregistered experiment ($n = 235$), we found that the value assigned to unchosen options is inversely related to their chosen counterparts. Moreover, this inverse relationship was associated with participants' memory of the pairs they chose between. Our findings suggest that deciding between options does not end the competition between them. Deliberation binds choice options together in memory such that the learned value of one can affect the inferred value of the other.

[1] Department of Psychology and Mortimer B. Zuckerman Mind, Brain, Behavior Institute, Columbia University, New York, NY, USA. [2] The Kavli Institute for Brain Science, Columbia University, New York, NY, USA. ✉email: natalie.biderman@columbia.edu; ds2619@columbia.edu

The Latin origin of the verb 'to decide' literally means 'to cut off' ('de' = off, 'caedere' = cut). The act of deciding is supposed to cut off the deliberation process to settle on one's choice. Yet, often the deliberation does not seem to end when a decision is made. Instead, the unchosen option continues to linger in one's mind. Over the years, research has shown that people continue to think about counterfactuals[1–6]. Yet, little is known about the mechanism that allows unchosen options to linger in our minds and the consequences of such lingering for the value of unchosen options.

Here we aim to address this gap. The starting point of our inquiry is that thoughts about unchosen alternatives do not come from thin air, but that they are instead tied to the chosen option[7–9]. For example, consider a decision you had to make, say about where to go on vacation. If the location you ended up choosing did not meet your expectations, this would of course lead you to devalue the choice you made[10]. But often in such situations, we also find ourselves automatically thinking back to the unchosen option, which we may now evaluate as a better option than it seemed at the time. Such post-decision experiences suggest that the options we deliberate between remain linked in our minds long after the decision was made. Indeed, deliberation is a comparative process, wherein options are evaluated simultaneously in relation to each other[11–13]. In this sense, deliberation provides a temporal and conceptual context shared by choice alternatives. Extensive memory research has shown that shared context creates an association between disparate elements, binding them to each other in memory[14–17]. These findings suggest that rather than 'cutting off' the unchosen option, deliberation may, paradoxically, tie the options together.

The possibility of choice options remaining associated in memory could have substantial consequences for how learning shapes value once the outcomes of a choice are revealed. Many studies have investigated the mechanisms by which outcomes of one's choice drive learning about the chosen option. According to prominent reinforcement learning models, learning the outcomes of chosen options leads to their value being updated based on the difference between the expected value and the actual experienced value[10].

The mechanisms of learning about unchosen options, however, are less clear. Studies of counterfactual learning have focused on how people learn from explicit information about what they could have gained if they had chosen the other option[18–24]. Yet, in most cases, people are not exposed to the outcomes of their forgone alternatives, leaving open questions about if, and what, people learn about unchosen options afterwards, when they experience outcomes of the choice that they made. In these situations, is the value of the unchosen options updated as well?

Here we hypothesize that the value of unchosen options is updated through their association with the chosen options. Central to this hypothesis is the role of memory. Recent advances suggest that memory associations can facilitate value inference and generalization. The basic idea is that once an association is formed, encountering one item leads to reactivation of the associated items[25–27]. In a recent study rats were required to make navigation decisions, and when they learned they made the wrong choice and their expected reward was nowhere to be seen, they reactivated the unchosen location[9]. Moreover, studies in humans have shown that if a reward is given to one of two associated items, the value of that reward can spread towards the associated items by reactivating a mnemonic network[28–30]. This reactivation mechanism has been shown to account for updating of chosen options, however, it remains unknown if it also affects the value of unchosen options.

The current study sought to determine whether the act of deliberation creates a memory association between the deliberated options and to explore the consequences of this association for later value learning. We hypothesized that learning about the outcomes of the choice leads to reactivation, in memory, of the unchosen option and this in turn leads to a change in the unchosen option's value. Unlike previous studies showing direct value transfer among associated items[28–30], choice options are associated within the deliberation context, which involves a contrast between the options. We, therefore, expect the value to transfer in the opposite direction. That is, if chosen items are explicitly rewarded (or unrewarded), we expect unchosen items to be inferred as unrewarded (or rewarded), a behavioral tendency we refer to as inverse inference of value. Thus, we hypothesize that deliberation may have a somewhat paradoxical role: While it is meant to dissociate choice options, deliberation binds them in our memory. When this bond is reactivated for the purpose of value updating, it continues to serve the deliberation goal of teasing the value of options apart.

Our prediction of an inverse transfer of value between choice options is based upon previous behavioral findings showing that participants continue to separate the value of options even after the decision was terminated. In studies of choice-supportive memory, participants exhibited a bias to better learn and remember the positive aspects of chosen options and the negative aspects of unchosen ones[31–34], thus increasing the contrast of value between the options. A similar contrast appears in studies of choice-induced preference change. These studies observed that after participants make a choice—even without any feedback on their choice—the value of chosen options tends to increase, while the value of the unchosen option tends to decrease[35–38]. Importantly, our study differs from, and therefore extends, choice-induced preference change studies in a crucial aspect. While choice-induced preference change studies examine how preferences are altered as a function of choice itself (regardless of the outcomes of such choices), the current study examines how the inferred value of unchosen options is updated as a function of outcomes.

To test our hypotheses, we devised a multiple-phase behavioral experiment in which participants deliberated between options and then learned the value of their choices (Fig. 1). First, we asked participants to deliberate between pairs of paintings and decide which painting would be more profitable in an upcoming auction (Phase 1, Deliberation; chosen items denoted as $S_{chosen}$ and unchosen items as $S_{unchosen}$). Importantly, we neither instructed participants to memorize the paintings nor did we specify that there was any dependency between the outcomes of paintings in each pair. Because we were interested in the effects of deliberation and memory on the valuation of unchosen options, we sought to verify that deliberation took place by asking participants to write down the reasons for choosing one option and not the other. Similarly, we sought to verify memory would be robust enough by repeating the deliberation trials several times and giving participants the chance to practice their decisions before committing to their choice.

In the second phase (Phase 2, Outcome Learning), participants learned about the outcome of each of their choices. Because we wanted to assess the effects of associative memory on value updating, we did not provide immediate feedback after each decision. Instead, after all decisions were made we presented only the chosen paintings alongside their auction outcomes.

In the next phase (Phase 3, Final Decisions) we sought to measure whether the outcome learning led to any value updating for either the chosen or unchosen options. We asked participants to make a new series of decisions between pairs of paintings, choosing the most-valuable painting in each pair. Each trial in this phase presented either two previously chosen stimuli (rewarded and unrewarded in the auction, denoted as

## Study Design

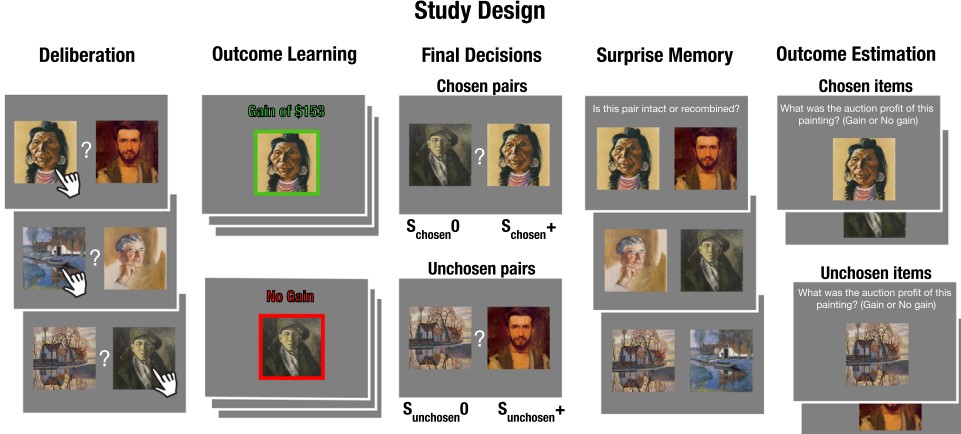

**Fig. 1 Study design.** In this multi-phase experiment, participants act as art dealers choosing paintings that are later sold in an auction. In the Deliberation phase, participants deliberate and decide which painting they would like to purchase to maximize their profits from the auction. After all decisions are made they learn the auction outcomes in the Outcome Learning phase. Only their selected paintings are presented alongside a colored frame and the profit they made in the auction: either gain (green frame) or no gain (red frame). Then, in the Final Decisions phase, participants are given a choice between two paintings and are asked to choose the most valuable. Unbeknownst to them, the decision pairs are either two previously chosen paintings (rewarded and unrewarded in the auction, denoted as $S_{chosen}+$ and $S_{chosen}0$, respectively) or two previously unchosen paintings (initially presented with $S_{chosen}+$ and $S_{chosen}0$, denoted as $S_{unchosen}+$ and $S_{unchosen}0$, respectively). The two pair types are randomly intermixed. In the Surprise Memory phase, participants are tested for associative memory of the deliberation pairs. They are presented with either the exact pairs from the Deliberation phase (intact) or pairs that include chosen and unchosen paintings that were not previously presented together (recombined). Experiment 1 included an Outcome Estimation phase, wherein participants are asked to estimate the auction profit of all paintings, including those they did not choose. Chosen and unchosen paintings are randomly intermixed. The study additionally included a pre-task and a post-task liking ratings of the paintings, detailed in the "Method" section. Stimuli were art images depicting people, objects, and scenes, randomly intermixed across conditions[77]. A subset of the stimuli are presented here. The art images in the figure include detailed images of the artworks PH-672 (1923), PP-241 (1936), and PH-269 (1941) by Clyfford Still, courtesy the Clyfford Still Museum, Denver, CO © 2021 City and County of Denver / ARS, NY, and the artworks Self Portrait (1900), A Farmbuilding (1900-1901), Farm Near Duivendrecht (1916) by Piet Mondrian, courtesy of Mondrian/Holtzman Trust © 2021. See Supplementary Fig. 1 for the full images.

$S_{chosen}+$ and $S_{chosen}0$, respectively), or two previously unchosen stimuli (presented earlier with $S_{chosen}+$ and $S_{chosen}0$, denoted as $S_{unchosen}+$ and $S_{unchosen}0$, respectively, Fig. 1). Participants were incentivized to choose the more valuable painting in order to earn extra bonus money based upon their performance.

To assess how memory of the deliberation impacts choices, after the last decision phase, participants were presented with a Surprise Memory Test (Phase 4) for the deliberation pairs. In Experiment 1, to assess explicit value inference of unchosen stimuli, at the end of the experiment we told participants that all paintings went on auction, including their previously unchosen paintings, and asked them to estimate which paintings were rewarded in the auction (Last Phase, Outcome Estimation). In addition, to control for participants' intrinsic preferences for any specific painting, before starting the main experiment we asked participants to rate each painting individually so that we could select items that were relatively neutral in their subjective value.

We were primarily interested in behavior on the Final Decisions phase and its relation to memory. Specifically, it is expected that if participants successfully learned the new values of the chosen items, then they should select $S_{chosen}+$ over $S_{chosen}0$. However, the critical question relates to their behavior on trials with unchosen items, for which they never received any direct feedback. As mentioned previously, we predicted an inverse inference of value for unchosen items. In the Final Decisions phase, where participants are asked to select the most-profitable items, this would result in the tendency to select $S_{unchosen}0$ over $S_{unchosen}+$. Finally, we hypothesized that this inverse decision bias should be larger when participants better remember which pairs of options appeared together.

Our main experiment (Experiment 1) was pre-registered on Open Science Framework (https://osf.io/chsvw) where we elaborated these hypotheses prior to data collection. To preview our results, across five distinct data sets (total $n = 612$), we found that participants were biased to select unchosen paintings previously paired with unrewarded paintings. This inverse decision bias was strongly related to their memory for the association between the options during deliberation.

## Results

All results were analyzed with Bayesian generalized linear models (see details in the "Analysis" section). In each model, we estimated a posterior distribution for regression coefficients and reported the median and 95% highest density interval (HDI) for coefficients of interest. If the 95% HDI of a coefficient excluded zero we concluded that the corresponding variable reliably predicted the outcome. Here, we refer to the value of chosen options after outcome learning as "learned value", since it results from participants' experience with the outcomes of their choices. In contrast, for the unchosen options that received no explicit feedback for learning, we use the term "inferred value".

**The value of chosen and unchosen options is inversely related.** We hypothesized that feedback about the chosen option would modulate the value of its associated unchosen option. To examine this, we focused on performance in the Final Decisions phase, allowing us to determine both direct outcome learning for chosen options (which were reinforced in the Outcome Learning phase) as well as inferences about the value of the unchosen options (which were not reinforced at any point in the experiment). We additionally examined an explicit outcome estimation report that participants gave for each painting at the end of the experiment.

Final Decisions. In Experiment 1 ($n = 235$, see preregistration on Open Science Framework: https://osf.io/chsvw), we found that when participants had to decide between two previously chosen items, they preferred $S_{chosen}+$ over $S_{chosen}0$ (probability to select

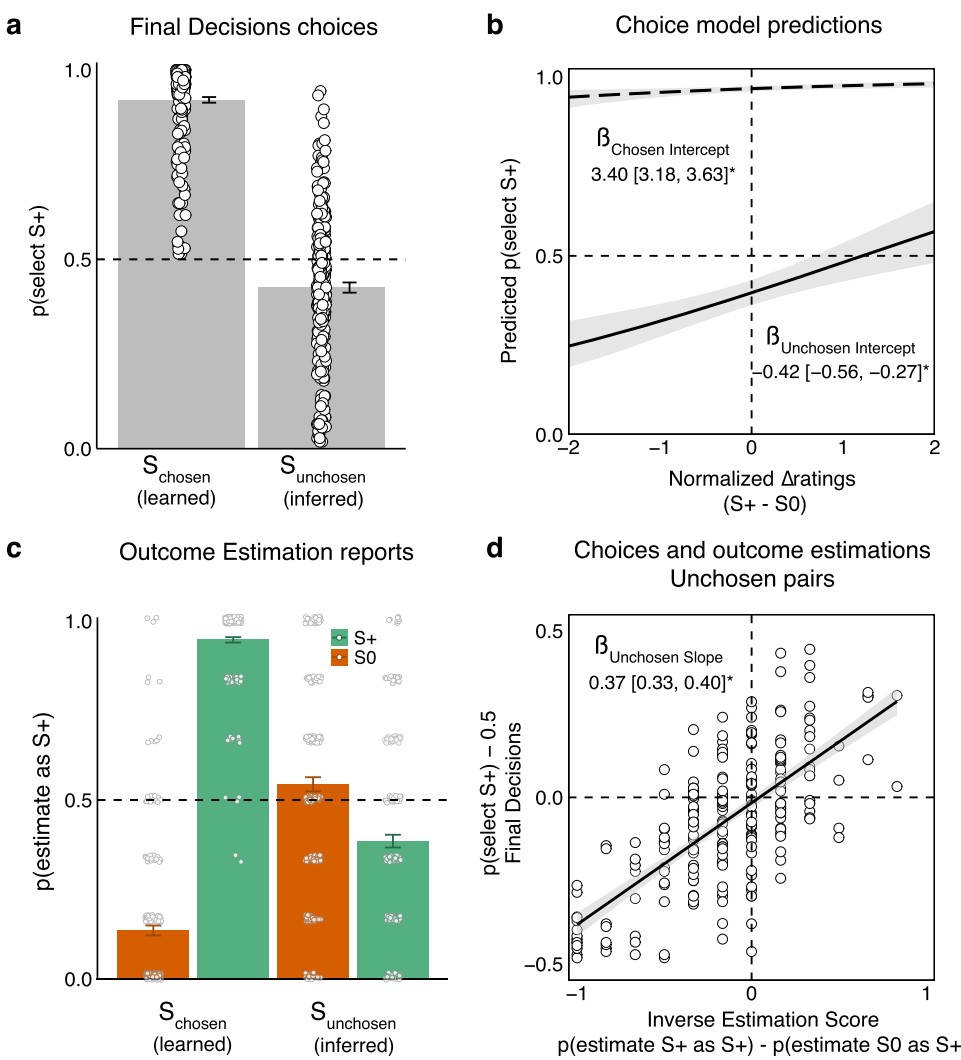

**Fig. 2 The inferred value of unchosen options is inversely related to the learned value of chosen options (Experiment 1, $n = 235$). a** When faced with a choice between two previously chosen stimuli, for which value was explicitly shown [$S_{chosen}$ (learned)], participants tended to select the rewarded option ($S_{chosen}+$), suggesting they successfully learned their values, while for pairs of previously unchosen options, which were never directly associated with any reward [$S_{unchosen}$ (inferred)], participants tended to select the option previously associated with an unrewarded item ($S_{unchosen}0$), demonstrating an inverse decision bias. **b** This inverse decision bias was observed even when controlling for initial subjective valuations of the choice options, in a Bayesian logistic regression predicting the probability to choose a rewarded item as a function of pair type and the difference in liking ratings. After rearranging the model coefficients, we can derive separate intercept terms for chosen and unchosen pairs. The intercept coefficient denotes the tendency to choose a rewarded item when there is no difference in liking ratings between the two choice options. For chosen pairs, the intercept is reliably positive, whereas for unchosen pairs it is reliably negative. **c** The inverse inference of value extends beyond the decision phase to explicit estimation of value. When asked to estimate the auction outcomes of each painting, participants correctly remembered the outcomes of the chosen paintings, yet showed the opposite pattern for the unchosen ones. **d** The tendency to select unchosen items previously paired with unrewarded items ($S_{unchosen}0$ over $S_{unchosen}+$) in the Final Decisions phase was correlated with an inverse estimation of value, i.e., the tendency to estimate $S_{unchosen}0$ as rewarded and $S_{unchosen}+$ as unrewarded in the Outcome Estimation phase. This relationship was assessed in a Bayesian linear regression predicting the mean probability to select rewarded items as a function of inverse estimation of value for chosen and unchosen pairs separately. In panels (**a**) and (**c**), error bars denote the standard error of the mean and points denote trial-averaged data of individual participants. In panels (**b**) and (**d**), the beta coefficients and model fits denote median and 95% highest density interval of the posterior distribution. In panel (**c**), green bars depict rewarded stimuli (S+) and orange bars depict unrewarded stimuli (S0) (for unchosen stimuli, this is the outcome of their chosen counterpart). Source data are provided as a Source Data file.

S+: 0.92 ± 0.01 [mean ± standard error], Fig. 2a), suggesting that they correctly learned the outcomes of their chosen items.

Critically, the opposite pattern was observed for unchosen items: participants were biased to select $S_{unchosen}0$ over $S_{unchosen}+$ (probability to select S+: 0.43 ± 0.01, Fig. 2a; for unchosen pairs, S+ items are paintings previously paired with a rewarded painting, and S0 items are paintings previously paired with an unrewarded painting). Recall that the participants never received explicit feedback on the unchosen items. We quantified these opposing patterns with a multilevel Bayesian logistic regression model that predicts the probability of selecting S+ items in the Final Decisions phase, conditional on the item being chosen or unchosen, and while accounting for the initial subjective likings of each painting. We computed a separate intercept term for chosen and unchosen items that reflects the tendency to select an S+ item when there is no liking difference between the choice items (see full model specification in the "Analysis" section, and coefficients in Supplementary Tables 2 and 3). A positive

intercept denotes a tendency to select S+ items and a negative intercept denotes a tendency to select S0 items. As expected, the estimated intercept for chosen pairs was positive (ß = 3.40 [3.18, 3.63]), while for unchosen pairs, it was negative (ß = −0.42 [−0.56, −0.27]; Fig. 2b).

Interestingly, we also observed that the tendency to choose unchosen paintings previously paired with unrewarded paintings ($S_{unchosen}0$) was accompanied with faster responses compared to choices of paintings previously paired with rewarded paintings ($S_{unchosen}+$, see Supplementary Text 1 and Supplementary Fig 2b). Considering the research showing that choices for rewards are faster than choices to avoid loss[39,40], this pattern in reaction times mirrors the inverse decision bias and suggests that participants viewed $S_{unchosen}0$ items to be more valuable.

Together, these results demonstrate that there is an inverse inference of value: in chosen pairs, participants tend to select S+ items, but in unchosen pairs they tend to select S0 items and this tendency is accompanied with faster reaction times.

Outcome Estimation. We next asked whether the aforementioned inverse decision bias extends to explicit change in value of choice options. To this end, we analyzed the explicit reports in the Outcome Estimation phase. Indeed, participants' explicit reports revealed that the estimated value of unchosen items mirrors the inverse decision bias (Fig. 2c). Participants estimated the auction outcomes of unchosen paintings to be inversely related to the chosen ones (probability to estimate an item as rewarded in $S_{unchosen}+$: 0.39 ± 0.02; and in $S_{unchosen}0$: 0.54 ± 0.02; multilevel Bayesian logistic regression: $ß_{unchosen}+ = −0.57$ [−0.75, −0.39], $ß_{unchosen}0 = 0.24$ [0.03, 0.45]). Here too, we validated that participants correctly learned which of their chosen paintings resulted in reward (probability to estimate an item as rewarded in $S_{chosen}+$: 0.95 ± 0.01; and in $S_{chosen}0$: 0.14 ± 0.01; multilevel Bayesian logistic regression: $ß_{chosen}+ = 3.74$ [3.30, 4.27], $ß_{chosen}0 = −2.64$ [−3.04, −2.30], see model specification in the "Analysis" section).

Lastly, we tested whether the inverse decision bias observed in the Final Decisions phase is related to participants' explicit reports in the Outcome Estimation phase. To assess overall performance in the Outcome Estimation phase, we computed an inverse estimation score for chosen and unchosen items separately. This was the difference in the mean probability to estimate an item as S+ between items that were rewarded (S+) and those that were not (S0, the difference between the green and red bars in Fig. 2c; for unchosen items the assigned outcome was that of the chosen item they were paired with). Negative scores will suggest an inverse inference of value (S+ items estimated as S0 and S0 items estimated as S+), and positive scores will suggest a direct inference of value (S+ items estimated as S+ and S0 items estimated as S0). Performance in the Final Decisions phase was computed as a linear transformation of the mean probability to select S+ items (i.e., p(select S+) −0.5, see the "Analysis" section for details), separately for chosen and unchosen pairs. Negative estimates signify a tendency to choose S0 over S+, and positive estimates signify a tendency to choose S+ over S0. We ran a Bayesian linear regression predicting choices in the Final Decisions phase as a function of choice type (chosen vs. unchosen pairs) and the inverse estimation score from the Outcome Estimation phase. We then rearranged the coefficients to get separate slope estimates of the effect of inverse estimation score on decisions for chosen and unchosen pairs. In both pair types, the slope term was substantial ($ß_{chosen} = 0.32$ [0.26, 0.37], $ß_{unchosen} = 0.37$ [0.33, 0.40], see Fig. 2d), suggesting that decision bias in the Final Decisions phase was correlated with the explicit estimation reports.

be related to the strength of the associative memory linking the two options. That is, to transfer value between a pair of options, participants need to have associated in memory the competing options included in each pair. To evaluate the relationship between decision bias and memory, we defined a summary statistic for each. We defined a pair memory score as the mean accuracy in the Surprise Memory phase, where participants were asked whether pairs of paintings appeared together in the Deliberation phase (see Supplementary Text 2 for analysis of the Surprise Memory phase). A decision bias score for each participant was calculated as the difference in the probability of choosing S+ paintings for chosen and unchosen pairs in the Final Decisions phase. Higher decision bias scores depict larger contrast in decision patterns between the two pair types, reflecting a stronger inverse decision bias. Bayesian linear regression between these two measures revealed that memory was associated with decision bias ($ß_{intercept} = 0.06$ [−0.08, 0.20]; $ß_{memory} = 0.67$ [0.47, 0.88]), such that better memory was related to stronger inverse decision bias across participants (Fig. 3a).

The correlation between memory and decision bias was also evident within participants. To assess within-participant variability, we used the outcome estimation reports of each participant to split their deliberation pairs into two kinds, per participant: (1) "direct transfer" pairs and (2) "inverse transfer" pairs. In the Outcome Estimation phase, for both chosen and unchosen items, we asked participants to judge whether an item results in a gain or no-gain. We defined direct transfer pairs as those for which participants judged both members of the pair in the same manner (judged both as gain or both as no-gain), alluding to the interpretation that the explicitly learned value of the chosen option was directly transferred to the unchosen option. We define the remaining pairs, i.e., those receiving opposed judgements, as inverse transfer pairs. We then compared pair memory performance between the two kinds of deliberation pairs. We found that pairs memory was more accurate for inverse compared to intact transfer pairs (mean ± SEM memory accuracy for inverse transfer: 0.75 ± 0.01; intact transfer: 0.68 ± 0.02; multilevel logistic regression predicting memory accuracy: $ß_{intercept} = 1.03$ [0.91, 1.16]; $ß_{pair-type} = 0.15$ [0.05, 0.25], Fig. 3b). Together, these findings suggest that memory is related to an inverse decision bias, both between- and within-participants.

Critically, our findings do not seem to be related to any explicit assumptions about the task structure. We examined the possibility that participants might have assumed that for every decision they make in the Deliberation phase, one painting would result in a gain and the other would not. While our instructions did not indicate any such relationship, we asked participants at the end of the study about their strategy in deciding between unchosen pairs. 22 out of 235 participants (9%) stated that they assumed that if their chosen painting did not gain in the auction, its associated unchosen painting probably did. The remaining participants stated they either guessed, chose according to their liking, or tried to look for similarities with the gaining chosen paintings. Importantly, the inverse decision bias remained substantial even when we excluded the 22 participants who explicitly used the inverse heuristic (probability to select $S_{unchosen}+$: 0.45 ± 0.01, unchosen intercept: ß = −0.30 [−0.44, −0.16], probability to select $S_{chosen}+$: 0.92 ± 0.01, chosen intercept: ß = 3.31 [3.09, 3.57]). Moreover, the inverse decision bias was still associated with outcome estimation score (unchosen pairs: ß = 0.35 [0.31, 0.40]) and memory of the initial deliberation pairs (ß = 0.56 [0.33, 0.78]).

**Decision bias is related to associative memory**. We hypothesized that the change in the value of the unchosen option would

**Replication of the main results across independent data sets**. We ran a series of follow-up experiments to test the robustness of

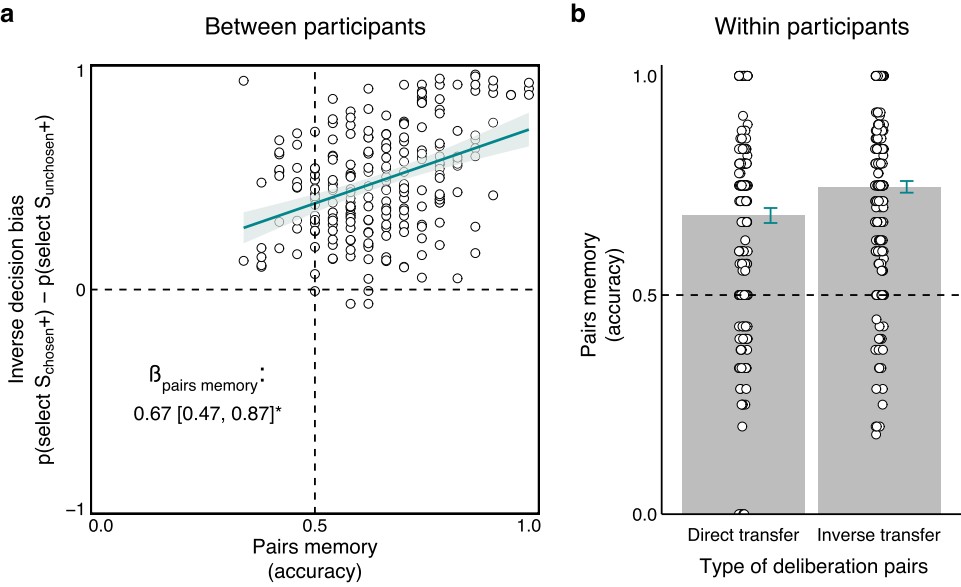

**Fig. 3 Inverse inference of value is related to associative memory (Experiment 1, _n_ = 235). a** The group-level inverse decision bias (difference in mean probability to choose rewarded items for chosen and unchosen pairs) is related to memory for the deliberation pairs. Points denote summarized observations of participants and the turquoise line denotes the fit of a Bayesian linear regression predicting inverse decision bias as a function of memory accuracy. Model fit and memory coefficient (beta) depict median and 95% highest density interval estimates. **b** The deliberation pairs were separated into two types for each participant based on outcome estimations (direct transfer: both the chosen and unchosen items within a pair were estimated with the same outcome, inverse transfer: the chosen and unchosen items were estimated to be in opposition). Pairs memory was better for inverse compared to direct transfer within participants. Points denote trial-averaged data of individual participants and error bars denote standard error of the mean. S+ denotes a rewarded stimulus (for unchosen stimuli, this is the outcome of their chosen counterpart). Source data are provided as a Source Data file.

our effects—namely, the inverse decision bias and its relation to memory—and their generalizability across variants of the experimental design. We ran three variants of the task with slight modifications in the Deliberation phase (see details in Supplementary Text 3 and Supplementary Fig 3), while keeping the general structure the same. We varied the number of times each pair was presented during the Deliberation phase (Experiment 2, _n_ = 96), the amount of gain that could be won for choosing the rewarded option (Experiment 3, _n_ = 95), and the high-level association between the pairs of paintings by informing participants that some of the pairs were painted by the same painter (Experiment 4, _n_ = 93). In all experiments, we replicated the inverse decision bias as well as its correlation with memory (see Fig. 4, Supplementary Text 3 and Supplementary Tables 1, 4 and 5 for details).

## Discussion

In five experiments (total _n_ = 612), we studied how exposure to outcomes of chosen options modulates the value of their unchosen counterparts. We discovered that after outcomes are revealed, the value assigned to unchosen options is inversely related to the learned outcomes of chosen options. This inverse bias manifested both in choice behavior and in explicit value estimation and it was associated with participants' memory of the pairs of options that they had deliberated about to begin with.

Our results have important implications for theories of value updating. Consistent with reinforcement learning models, we show that outcomes change the value assigned to chosen options. With respect to unchosen options, previous studies showed that direct feedback about their hypothetical outcomes leads to changes in value and subsequent behavior[18,20–22,24,41–43]. In life outside the laboratory, however, we are usually exposed only to the outcomes of our chosen option. Here we show that such ecological exposure is sufficient to facilitate value inference even

for unchosen options, and that this value inference is related to memory.

We found that stronger memory for the deliberated options is related to a stronger discrepancy between the value assigned to the chosen and unchosen options. This result suggests that choosing between options leaves a memory trace. By definition, deliberation is meant to tease apart the value of competing options in the service of making the decision; our findings suggest that deliberation and choice also bind pairs of choice options in memory. Consequently, unchosen options do not vanish from memory after a decision is made, but rather they continue to linger through their link to the chosen options.

We show that participants use the association between choice options to infer the value of unchosen options. This finding complements and extends previous studies reporting transfer of value between associated items in the same direction, which allows agents to generalize reward value across associated exemplars. For example, in the sensory preconditioning task, pairs of neutral items are associated by virtue of appearing in temporal proximity. Subsequently, just one item gains feedback— it is either rewarded or not. When probed to choose between items that did not receive feedback, participants tend to select those previously paired with rewarded items[28–30,44]. In contrast, our participants tended to avoid the items whose counterpart was previously rewarded. Put in learning terms, when the chosen option proved to be successful, participants' choices in our task reflected avoidance of, rather than approach to, the unchosen option. One important difference between our task and the sensory preconditioning task is the manner in which the association is formed. In both tasks a pair of items appears in close temporal proximity, yet in our task participants are also asked to decide between these items and the act of deliberation seems to result in an inverse association between the deliberated options.

Another interesting difference between the current findings and previous findings of value inference is related to whether the

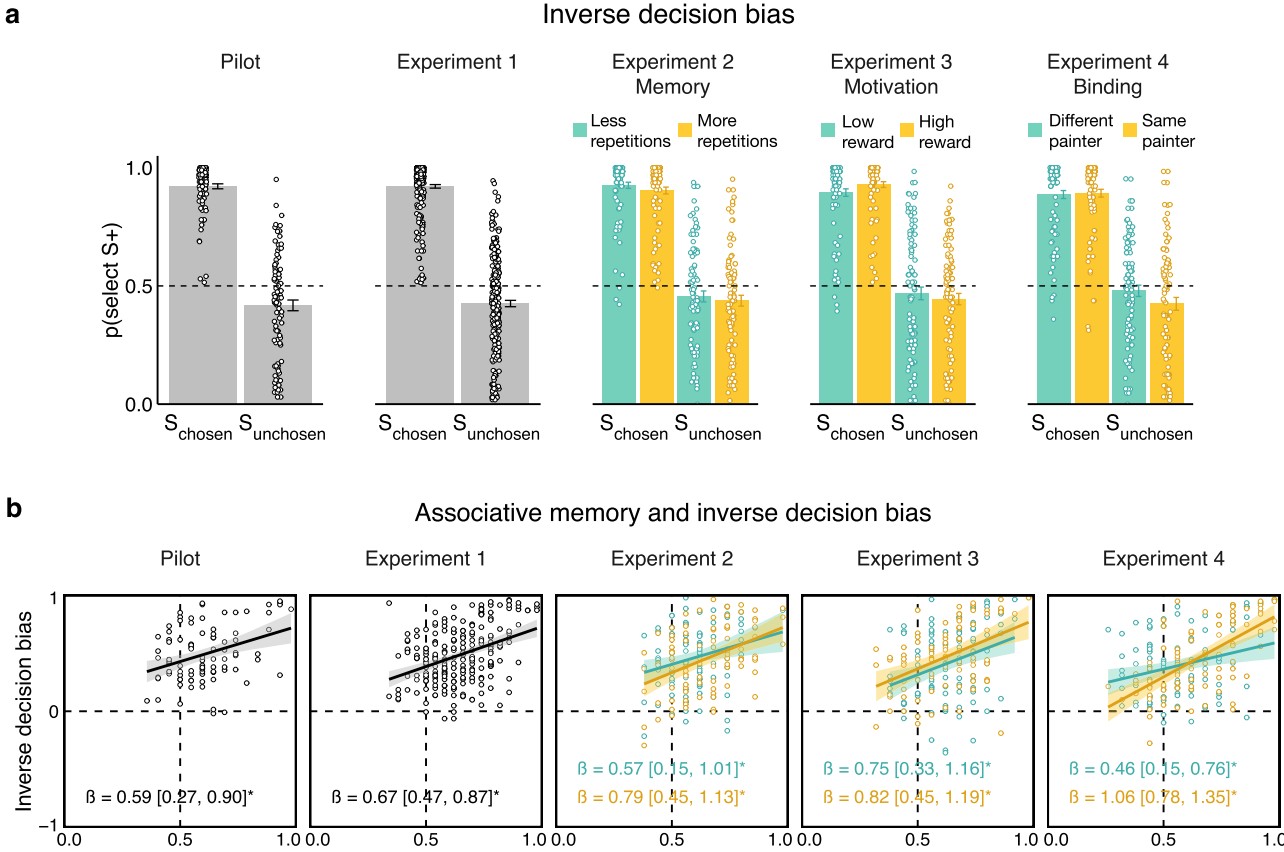

**Fig. 4 Replicating the main effects across five distinct data sets.** Experiment 1 ($n = 235$) was pre-registered on Open Science Framework (https://osf.io/chsvw) following a Pilot study ($n = 93$). Experiments 2 to 4 include different experimental manipulations of the Deliberation phase (depicted by yellow and cyan colors). We manipulated the number of times each deliberation pair was presented (Experiment 2, memory manipulation, $n = 96$), the magnitude of potential gains (Experiment 3, motivation manipulation, $n = 95$), and the high-level association of pairs by assigning some of them as belonging to the same painter (Experiment 4, binding manipulation, $n = 93$). **a** The tendency to select rewarded paintings for chosen pairs ($S_{chosen}+$), but unrewarded ones for unchosen pairs ($S_{unchosen}$0) was replicated across conditions. **b** Memory of the deliberation pairs was related to inverse decision bias (difference in decision tendency of chosen and unchosen pairs) in all experiments and conditions. Points denote trial-averaged data of individual participants. In panel (**a**), error bars denote standard error of the mean. In panel (**b**), betas represent the memory slope coefficients in a Bayesian linear regression predicting inverse decision bias as a function of memory accuracy, and for both betas and model fits we show median and 95% highest-density intervals of the corresponding posterior distribution. S+ denotes a rewarded stimulus (for unchosen stimuli, this is the outcome of their chosen counterpart). Source data are provided as a Source Data file.

memory associations driving changes in value are implicit or explicit. In sensory preconditioning and similar tasks (such as acquired equivalence[25]), memory's influence on value-based decisions appears to happen outside of conscious awareness and explicit memory for associations is orthogonal to their influence on choice. Here we find, repeatedly, that explicit memory for the association between choice options is related to the extent of the inverse bias in decisions. An important challenge for future work will be to develop a principled understanding of the factors that determine when this relationship is implicit vs. explicit and to what extent it may be related to whether the initial encoding of associations is incidental (as in sensory precondition) or whether it is a deeper encoding that itself happens in the context of a decision (as in the current task).

In our data, we observed large individual differences in value transfer from chosen to unchosen options (see Fig. 2a). While the majority of participants transferred value in the opposite direction, some exhibited a direct value transfer. This tendency was evident even between deliberation pairs of the same participant. Notably, individual differences in value transfer is not uncommon in the sensory preconditioning task[28–30,44], but future studies could use additional physiological and neuroimaging methods to

find correlates that predict these differences. An additional possible source of variation could be related to retroactive or proactive interference, which may lead to the chosen and unchosen options becoming incorrectly associated. This cannot account for the full effects we observe and, in particular, does not account for the observed relationship between memory and decision bias. Nonetheless, in the future it will be interesting to assess how much of the variance we observe in decision bias is related to forgetting and proactive/retroactive interference.

The inverse inference of value resonates with studies showing that making a decision leads to selective memory for information supporting the decision. The classical finding is that participants remember their chosen option more favorably and remember their unchosen option less favorably[31,32,34], and they also exhibit higher learning rates for the positive outcomes of their chosen options and the negative outcomes of their unchosen options[33,41]. These tasks differ from our task in that participants are exposed directly to the outcomes of their unchosen options and exhibit biases in the memory of these outcomes. Here we show that even without any exposure to the unchosen option's outcome, participants infer that its value is opposed to that of the chosen option. Importantly, the inverse inference of value does not always

support choices: unchosen options could be inferred as rewarding if the chosen option was not rewarding.

Our findings suggest that the act of deciding can modify an option's value, consistent with studies of choice-induced preference change[35,36,38,45–47]. These studies show that for two equally valued options, the mere act of deciding induces a tendency to overvalue the chosen option and undervalue the unchosen one. Both options' values are modified right after the decision is made, as a by-product of it. Our study differs from, and therefore extends, choice-induced preference change studies in an important way. In choice-induced preference change, the change in value occurs as a function of a single choice and a choice alone, not as a function of learning from subsequent outcomes. The latter is what our task targets. While our study did not evaluate changes in subjective preferences (e.g., using ratings), but rather changes in the value people assign to choice options after learning about their outcomes, our results echo choice-induce preference-change effects. As with choice-induced preference-change, we find that people tend to separate the value of their choice options after the decision was terminated. Yet, importantly, we find that people use the outcomes they received for their chosen options as a reference point to infer the value of their unchosen options. They do not only devalue these options, but in some cases they might even assume they were better options than their chosen ones. Finally, our findings are also consistent with recent studies showing that choice-induced preference change effects were correlated with a better memory of the initial choice[35,48–51].

Our findings add to recent advances showing that episodic memory guides value-based decisions (for a review see ref. [52]). Recent studies have found that people consult specific past events when evaluating their next choice[53,54]. Decision makers are biased to sample a past decision when given an incidental reminder of the decision context[55], if the option resulted in a high reward and they remember the association between the option and the reward[56], and when the option seems familiar[57]. Furthermore, recent studies have shown that the simple act of making a decision, rather than having someone else make it for you, increases the episodic memory of that decision[50,58,59], thereby making it more accessible for later retrieval. The current study both builds upon and extends these findings by suggesting that memory mechanisms also contribute to the valuation of unchosen options, potentially by retrieving the deliberation context in which the choice options were associated.

An important future direction will be to test whether memory and deliberation are necessary for inverse bias to occur. The present study was designed to assess the effects of deliberation, and memory of the deliberation, on the valuation of unchosen options. To facilitate this intent, we designed the task to increase the chances that participants would actually deliberate and would be able to learn the deliberation pairings as well as the outcomes of the chosen options. Following this logic (and prior studies[28]), we repeated the pairings and outcomes several times to assure they were learned sufficiently, and asked participants to write down the reasons for choosing one option and not the other to promote deliberation. Furthermore, we did not provide immediate feedback after every decision but rather delayed the feedback and presented it only for the chosen options. This design feature was meant to facilitate the use of long-term memory associations formed during deliberation to transfer value between choice options, rather than doing so while the two options are kept simultaneously in working memory when the feedback is immediate.

As a next step, having found that sufficient deliberation and memory reliably manifest in inverse inference of value, it will be important to further determine the boundary conditions necessary for this effect. For example, when deliberation is not as extensive or when outcomes are provided immediately after the decision was made (as in standard reinforcement learning tasks), rather than later on. In Experiment 2, we manipulated the memory strength of the deliberation pairs by changing the number of times they were repeated (two vs. six times). Pairs that were repeated more often were remembered better than those repeated less often. Critically, however, there was no difference in decision bias for unchosen pairs between these conditions, nor was there any difference in the correlation of decision bias with memory (see Supplementary Text 3). Future studies could assess the parameters that modulate the inverse inference of value.

In addition, while the decisions in our study involved deliberating and choosing between two options, an open question is what would happen in non-binary decisions. For example, when the decision involves more than two options[60], when the decision is between engaging with an option or continuing to search for better alternatives (foraging decisions)[43,61–63], or when the decision requires to first select the options to consider amongst many alternatives[64], among other types of decisions. Future studies could assess whether learning about chosen options in non-binary decisions modulates the value of all unchosen alternatives considered in a specific context, only the one dominant unchosen alternative, or none at all. These different possible outcomes would provide insight on how people reason about the interplay between the value of chosen and unchosen options across different contexts.

Our study sheds light on value transfer between chosen and unchosen options and raises questions about when the value transfer takes place. We eliminated the possibility of value transfer during deliberation (or immediately following it) by withholding any feedback at the initial deliberation stage. For the value of unchosen options to change in relation to the chosen ones, participants had to associate the two options to each other in memory. We, therefore, suggest that deliberation creates a memory association between choice options, and this association is later reactivated to facilitate value transfer. Previous studies have suggested that value can transfer during encoding[28,29] (here, when participants learn the value of chosen options during Outcome Learning phase), or right before new decisions are faced[44,65] (here, in the Final Decisions phase). While future studies are needed to directly disambiguate these options, our findings provide initial evidence supporting the hypothesis of value transfer during encoding. Specifically, we found that responses were faster when choosing unchosen items previously paired with unrewarded items ($S_{unchosen}0$), suggesting that these items were perceived as more valuable[39,40]. These findings allude to the possibility that the positive value of $S_{unchosen}0$ items was most likely already constructed before the participants faced the new choice in the Final Decisions phase; if the value was constructed on the spot during the decision phase, we would have expected slower, not faster, reaction times.

An important open question is whether the pattern of inverse bias is specifically modulated by counterfactual thinking. That is, when participants learn about the outcomes of their decisions, do they engage in thinking about what would have happened had they chosen the other, unchosen, option, and consequently infer that the value of that option is in opposition to their chosen option. Studies of counterfactual thinking suggest that thoughts about counterfactuals may arise when encountering outcomes of chosen options[7]. Furthermore, recent studies have shown that counterfactual thinking activates similar brain regions as episodic memory[66,67], and counterfactual construction can lead to memory distortions[68].

Relatedly, spontaneous counterfactual thoughts are triggered more frequently after experiencing negative outcomes compared

to positive ones[7,8,69,70]. Our behavioral test (Final Decisions phase) had participants choose between rewarded and unrewarded stimuli in chosen and unchosen pairs of stimuli, so we could not discern whether the tendency to select $S_{unchosen}0$ over $S_{unchosen}+$ stemmed from the inferred positive value of $S_{unchosen}0$, the inferred negative value of $S_{unchosen}+$, or both. When participants were asked to explicitly report the potential outcomes of unchosen stimuli (Outcome Estimation phase), there was a greater tendency to report an inverse outcome for $S_{unchosen}+$ compared to $S_{unchosen}0$ (see Fig. 2c). That is, unchosen stimuli that were paired with rewarded chosen stimuli were categorized more frequently as unrewarded. Yet this tendency might have originated from a weaker signal for unrewarded chosen stimuli compared to rewarded ones, because we did not introduce losses but only zero gains. Future studies could introduce losses rather than no gains and test whether the asymmetry following negative versus positive outcomes in counterfactual thinking studies also translates to asymmetry in the modulation of the value of unchosen options.

To conclude, our findings suggest that deciding between competing options does not end the competition between them. Deliberation binds choice options together in memory, such that the learned value of one can affect the inferred value of the other.

## Methods

**Preregistration and sample size determination**. Experiment 1 was preregistered on Open Science Framework (https://osf.io/chsvw). Prior to data collection we ran a Pilot study ($n = 93$) which confirmed our main hypotheses and guided later sample size decisions. Specifically, we found that participants exhibited an inverse decision bias (selected rewarded items in chosen pairs and unrewarded items in unchosen pairs), which was correlated with pairs memory (see Fig. 4 and Supplementary Tables 1–5). The goal of Experiment 1 was to replicate these findings using a substantially larger sample size. Following[71], we defined the sample size of Experiment 1 to be 2.5 times the sample size of the Pilot study, which is 235 participants (we calculated 2.5 times the original sample size of 93 participants and rounded up by a few). A sample size of 235 participants gave us above 99% power to detect the effect size of the Pilot study, computed based on a simplified version of our main logistic regression model (see the "Analysis" section). Namely, we fitted a logistic function for each participant using "lme4" package[72] and then tested the unchosen intercept coefficients against zero ($t(92) = -3.71$; $p < 0.001$; Cohen's $d = 0.39$; power analysis was performed using "pwr" package[73] and Cohen's d computation was performed using "lsr" package[74]).

**Participants**. The research was approved by the Institutional Review Board (IRB) at Columbia University through Columbia IRB Protocol #AAAI1488. 235 Mechanical Turk participants took part in Experiment 1 for an average payment of $6 (mean age: 28.54 ± 4.49, 132 female, 101 male, 2 other). All participants provided informed consent for their participation in the experiment. 377 participants participated in the Pilot study and Experiments 2 to 4 (see Supplemental Text 3 for demographic information). Additional 50 participants across all experiments were removed from analyses because they met our predefined exclusion criteria for online studies (see below, all exclusion criteria were preregistered on OSF: https://osf.io/chsvw). We added a restriction on Mechanical Turk to include only US-based participants within the age range of 18–36, and an approval rate of above 90%.

**Exclusion criteria**. We applied the following exclusion criteria which were all aimed to ensure that the online participants were attending the task: (1) Below chance performance (probability to choose gain items below 0.5) in the Final Decisions phase for chosen pairs, indicating participants who did not learn the new values of chosen paintings (for Experiments 2 to 4, the exclusion performance is computed across conditions); (2) More than 25 missed responses in either the Outcome Learning phase (where participants had to register the outcomes they observed) or the Final Decisions phase; (3) More than 25 events where participants were browsing a different window in any experimental phase (blur-focus events detected using jsPsych library[75]); (4) More than 10 trials in the Deliberation phase where responses were too fast (below 300 ms; these trials were accompanied with a warning), signifying no actual deliberation; and (5) More than 10 failed attempts to answer a comprehension quiz administered after instructions in any experimental phase. Participants who met at least one of these exclusion criteria were removed from all analyses.

**Materials**. Stimuli were images of representational paintings that depict people, landscapes, or objects, randomly intermixed across conditions. The stimuli were

collected by Celia Durkin from various online databases, and a subset of these stimuli are published online[76]. The stimuli were converted to $300 \times 300$ pixel size and were presented on a gray background (RGB: 128, 128, 128; see Fig. 1).

**Procedure**. Experiment 1 included the following consecutive phases: (1) Pre-task Ratings, (2) Deliberation, (3) Outcome Learning, (4) Final Decisions, (5) Surprise Memory, (6) Post-task Ratings, and (7) Outcome Estimation. Each phase began with instructions followed by a comprehension quiz. Participants were not informed about the upcoming phases.

*Pre-task ratings*. Participants were presented with 60 paintings and were asked to rate their liking of each painting using a continuous scale (from "not at all" to "very much", responses were then scaled from 0 to 100). At the end of the phase, the paintings were sorted by their ratings, and the 24 paintings rated in the middle of the distribution were selected to construct the deliberation pairs. The selected paintings were shuffled and distributed across 12 pairs.

*Deliberation*. Participants were instructed to act as art dealers who will choose paintings to sell in an auction and their goal is to maximize profits from that auction. During the Deliberation phase, participants saw 12 pairs of paintings and were asked to deliberate and choose one of the paintings in each pair. They were instructed to take their time and were given up to 10 s to make a decision. If they did not respond during this time, they were prompted to make the decision again and to respond more quickly. To ensure deliberation and encoding of the pairs, we told participants they will practice the decisions a couple of times before committing to their final choices. The 12 deliberation trials were repeated three times in three separate blocks, with random order of trials within each block. Overall, participants were consistent with their initial choices (mean and standard error of the number of times participants repeated the same decision: 2.87 [0.01]). In subsequent phases, we used the choices made in the last Deliberation block. To increase deliberation, participants were asked to write down the reasons for their decisions using a text box. The text boxes appeared once for every deliberation pair across the first two blocks. Participants were told that one of their decisions will be played out for real, and they will receive 1% of their chosen painting's auction earnings, if indeed it resulted in a gain. At the end of the Deliberation phase, all participants were informed that they received extra $1.5 bonus money (we made sure one of the chosen items gained $150).

*Outcome learning*. Because we were interested in the long-term effects of associative memory on value updating, we did not provide immediate feedback after every decision. We assumed that after making a choice, participants could still maintain both choice options in their working memory and feedback for the chosen option could transfer to the unchosen option just by virtue of being activated simultaneously in working memory. Thus, only after the completion of all decisions, we presented the outcomes for chosen paintings alone. Half of the deliberation pairs were randomly assigned as rewarded items (denoted as S+), with earnings centered around $150 and standard deviation of $10. The other half were assigned as unrewarded items, with $0 earnings (denoted as S0). Paintings were presented in the center of the screen alongside a colored outcome and frame (green for S+, red for S0, see Fig. 1) for 2 s. To facilitate learning, we repeated each painting six times across three blocks, in a randomized order. To ensure participants' attention, we asked them to press the space bar to see the outcome auction and then to register the outcome by pressing a corresponding key while it is presented on the screen (up arrow for gain, down arrow for no gain). If they missed an outcome registration they saw a warning asking them to respond faster.

*Final decisions*. Participants were asked to prepare a portfolio of high-valued paintings and to do so they had to make additional decisions. They were incentivized to select the more valuable painting in each pair by being informed that they could earn extra bonus money based upon their performance in this phase. The final-decisions pairs were constructed from all possible combinations of S+ and S0 paintings, separately for previously chosen and unchosen paintings (for unchosen paintings, S+ and S0 assignment was according to their chosen counterparts). This yielded 36 unique chosen pairs and 36 unique unchosen pairs, for a total of 72 pairs. To increase the number of trials, we repeated the decision trials three times across three separate blocks, each included all 72 pairs, randomly intermixed. In each block, the rewarded painting appeared on the left and the right sides equal number of times. Participants had 2.5 s to make a decision and, if they failed to respond during this time, they were shown a warning asking them to respond faster. Unbeknownst to participants, the potential bonus money was based only on their performance for the chosen pairs and was up to $2.

*Surprise memory*. To test for memory of the deliberation pairs, we presented 24 pairs of paintings and asked participants whether each pair was intact or recombined. Intact pairs were pairs of paintings that appeared in the Deliberation phase, and recombined pairs were pairs that included a chosen painting and an unchosen painting that did not previously appear together. If participants responded "intact", they were also asked to indicate which of the two paintings they previously had chosen in the Deliberation phase.

*Post-task ratings.* Similarly to the Pre-task Ratings phase, we presented all 60 paintings again and asked participants to rate how much they like them, using a continuous scale.

*Outcome estimation.* On this final phase, we told participants that all paintings were sent to auction, including those they didn't choose in the beginning of the experiment, and asked them to estimate the auction outcomes. We presented previously chosen and unchosen paintings in a random order, and for each painting we asked participants whether the painting resulted in a gain or not, and then to rate how confident they are in their response from 1 ("completely unsure") to 6 ("completely sure").

At the end of the experiment we asked participants about their decision strategies throughout the task. We were specifically interested to examine the possibility that participants might have adopted an inverse heuristic. That is, they might have assumed that for every decision made in the Deliberation phase, one painting would result in a gain and the other would not. To this end, we first asked participants what was their decision strategy in the Deliberation phase and then we asked them how they decided between pairs of paintings for which they received direct feedback (chosen pairs) and for which they received no feedback (unchosen pairs) in the Final Decisions phase.

**Analysis.** All data and analysis codes are publicly available on GitHub[77]. Data analysis was performed in R[78] using RStudio[79]. All results were analyzed with generalized linear models using the "rstanarm" package[80], which performs approximate Bayesian inference over the regression coefficients. Instead of a maximum-likelihood procedure that provides a single point-estimate for each coefficient, Bayesian inference targets the full posterior distribution of each coefficient, which combines our prior assumptions and the observed data:

$$p(\beta|x) = \frac{p(\beta)p(x|\beta)}{p(x)} \qquad (1)$$

In our models, the denominator of this expression is intractable to compute, preventing an exact solution for the posterior distribution. Therefore, we approximate it by taking samples from it, through a procedure called Hamiltonian Monte Carlo. We improve the fidelity of our samples by running multiple independent sampling processes ("chains"). For every model, we used default priors and ran six chains with 4000 iterations each (2000 iterations per chain were used as warm-up). To determine convergence of each chain, we made sure that for all model coefficients the effective sample size of simulation draws was >900 and the R-hat statistic was around 1.0[81,82]. To evaluate our effects, for each regression coefficient of interest, we report the median of the posterior samples and their 95% HDI. Since a regression coefficient of zero indicates no relationship between a predictor and an outcome, we determined that a variable reliably predicts an outcome only if its 95% HDI excluded zero.

Moreover, when possible, we ran multilevel (or hierarchical) models, in which a participant's regression coefficient is drawn from a group-level coefficient distribution. Such an approach is more robust to outlier participants and observations. The parameters of this group-level distribution are of special importance, since they indicate whether an effect is present or not across all participants. In all multilevel models, all predictors varied by participants.

*Analysis of choices in Final Decisions phase.* We ran a multilevel Bayesian logistic regression that predicted the probability of choosing S+ over S0 as a function of (1) choice condition (chosen or unchosen pairs, centered predictor), (2) the difference in pre-task liking ratings between the items in each pair (S+ minus S0), and (3) their interaction. We first define the sigmoid function whose output is bounded between 0 and 1:

$$\sigma(x) = \frac{1}{1 + e^{-x}} \qquad (2)$$

And we model the probability of selecting S+ as

$$p(\text{select } S+) = \sigma(\beta_0 + \beta_1 * \text{choice} + \beta_2 * \Delta\text{ratings} + \beta_3 * \text{choice} : \Delta\text{ratings}) \qquad (3)$$

We normalized each participant's ratings to control for differences in the overall use of the scale using a z-score normalization (i.e., some participants tend to use higher/lower values of the continuous scale), and then subtracted the normalized rating of the unrewarded painting (S0) from the rewarded painting (S+) in each decision trial. Trials where participants missed a response were excluded from analyses.

Notably, by design, chosen and unchosen pairs involve different value signals to guide choices; for chosen pairs the outcomes were presented explicitly, whereas for unchosen pairs they can only be inferred. We, therefore, expected the choice coefficient to be substantial (suggesting a difference in decision tendency in chosen versus unchosen pairs), even if participants chose randomly in trials of unchosen pairs. Consequently, we rearranged the equation, effectively decomposing our model into two separate regression models, one for chosen items and one for unchosen ones. Each such model is predicting the probability of gain choice as a function of ratings. We do so by computing a separate Δratings coefficient and a separate intercept term for chosen and unchosen items. The Δratings coefficient is a slope term indicating the influence of ratings on choice (computed as ß₂ +

ß₃*choice, where choice = 1 for chosen pairs, and choice = −1 for unchosen pairs), and the intercept term quantifies the tendency to choose rewarded items when there is no difference in ratings (computed as ß₀ + ß₁*choice). Our prime measure of interest was the intercept term.

*Analysis of outcome estimation.* To test whether participants explicitly infer that unchosen paintings are valued in opposition to chosen paintings, we analyzed outcome estimation reports. We ran a multilevel logistic regression predicting the probability to estimate an item as S+ as a function of (1) choice (chosen or unchosen item, centered predictor), (2) actual auction outcomes (S+ or S0, for unchosen items this is the outcome of their associated chosen items, centered predictor), and (3) their interaction.

$$p(\text{estimate } as\ S+) = \sigma(\beta_0 + \beta_1 * \text{choice} + \beta_2 * \text{outcome} + \beta_3 * \text{choice} : \text{outcome}) \qquad (4)$$

To evaluate the probability that items were estimated as rewarded in a specific condition (e.g., unchosen items previously paired with rewarded items, i.e., $S_{unchosen}+$), we plugged in the relevant predictors and summed the coefficients (e.g., for $S_{unchosen}+$, choice = −1, outcome = 1, therefore ß = ß₀ + (−1)*ß₁ + 1* ß₂ + 1*(−1)*ß₃).

*Analysis of the relationship between decision bias and outcome estimations.* To test whether choices in the Final Decisions phase are predicted by outcome estimations, we computed an inverse estimation score for chosen and unchosen items separately. This is the difference in the mean probability to estimate an item as S+ between items that were rewarded (S+) and those that were not (S0, for unchosen items, the assigned outcome was that of the chosen item they were paired with). Negative scores signify an inverse inference of value, i.e., S+ items estimated as S0 and S0 items estimated as S+. Performance in the Final Decisions phase was measured as the mean probability to select S+ items (over S0), separately for chosen and unchosen pairs. To allow interpretability of the intercept terms in this model, we had to make sure that 0 value had a significant meaning in our analysis. Accordingly, we subtracted 0.5 from the mean probability to select S+ items. Negative estimates signify a tendency to choose S0 over S+, and positive estimates signify a tendency to choose S+ over S0.

We ran a linear regression predicting the probability to select S+ as a function of (1) choice (chosen or unchosen pairs, centered predictor), (2) inverse estimation score, and (3) their interaction. Because we computed inverse estimation scores separately for chosen and unchosen items, we could assign these scores to their respective pairs in the Final Decisions phase.

$$p(\text{select } S+) = \sigma(\beta_0 + \beta_1 * \text{choice} + \beta_2 * \text{estimation score} + \beta_3 * \text{choice} : \text{estimation score}) \qquad (5)$$

We then rearranged the regression coefficients to effectively get two regression models, one for chosen pairs and the other for unchosen pairs. Each model is predicting the probability to select S+ as a function of inverse estimation score. For every model, we computed an intercept term ($\beta_0 + \beta_1 * \text{choice}$), quantifying the mean probability to select S+ items when there is no difference between estimation scores, and a slope term ($\beta_2 + \beta_3 * \text{choice}$), quantifying the influence of inverse estimation score on choices. Our measure of interest was the slope term.

*Analysis of associative memory and inverse inference of value across and within participants.* We first assessed the relationship between associative memory and inverse decision bias across participants. To this end, for each participant, we computed two measures. The first was average accuracy in pairs memory responses collected during the Surprise Memory phase. The second was a decision bias score, measuring the contrast in decision tendency between chosen and unchosen pairs. This was operationalized as the mean probability to choose a rewarded item in chosen pairs minus unchosen pairs. Values closer to 1 signify a large inverse decision bias effect. We used a Bayesian linear regression to predict inverse decision bias as a function of memory. The coefficient of interest in this model is the memory slope term.

To assess the relationship between associative memory and inverse estimation of value within participants we used responses in the Outcome Estimation phase, where participants were asked to estimate the auction outcomes of all paintings. This allowed us to retroactively separate the deliberation pairs of each participant into two pair types: (1) direct transfer—both the chosen and unchosen paintings within a pair were later estimated with the same outcome, or (2) inverse transfer—the two paintings were later estimated with the opposite outcome. We then wanted to assess whether there was a difference in associative memory of these two pair types. To this end, we ran a multilevel logistic regression predicting memory accuracy from the Surprise Memory phase as a function of pair type (direct vs. inverse transfer). For ease of analysis, we only used responses from the "intact" trials where pairs appeared as they were presented during the Deliberation phase.

**Reporting summary**. Further information on research design is available in the Nature Research Reporting Summary linked to this article.

## Data availability
Behavioral data generated and analyzed during the current study are available in GitHub[77] with the identifier (data https://doi.org/10.5281/zenodo.4926569). Source data are provided with this paper.

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

## Acknowledgements

We thank Dan Biderman for insightful discussions and for providing feedback on an earlier draft. We also thank Celia Durkin for comments on an earlier draft and the Shohamy lab for helpful conversations on this project. We are grateful for funding support from the NSF (award # 1822619), NIMH/NIH (award # MH121093), and the Templeton Foundation (grant #60844).

## Author contributions

N.B. and D.S. designed the experiments. N.B. programmed the experiments, collected the data and performed data analysis. N.B. and D.S. wrote the manuscript.

## Competing interests

The authors declare no competing interests.
