## [Peer Review File · Nature Communications]

REVIEWER COMMENTS

Reviewer #1 (Remarks to the Author):

This manuscript describes a series of behavioral studies in which subjects deliberated between pairs of artwork and then selected the piece to “send to auction.” In a subsequent phase, those selected pieces—but not the unselected pieces—were associated with outcome (either gaining money at auction, or not gaining money). In the next phase, subjects were then shown pairs of previously chosen or previously unchosen pictures and made a decision of which piece to add “to their portfolio.” For the previously-chosen pairs, one pair had gained money at auction (+) and one had not (0), so there was a straightforward choice—and subjects readily chose the pieces that had won money. For the unchosen pairs, one piece had previously been paired with a piece that had gained money (+) and the other piece had previously been paired with a piece that did not gain money (0). Here, there is a more interesting question of which piece subjects would choose. Across several studies, there was a clear and consistent tendency for subjects to systematically choose the piece that had previously been associated with a piece that did NOT gain money (and this was related to subjects’ memory for which pieces had originally occurred together). In some ways this is surprising as one might predict that a positive outcome for a chosen piece would transfer to the unchosen piece. But, the result is exactly the opposite (what the authors call inverse inference). The finding is highly reminiscent of—and consistent with—the literature on choice supportive memory (where there is a bias to remember chosen options more favorably and unchosen options less favorably). In fact, the biggest concern with the current manuscript is whether it is a sufficient advance beyond the choice supportive memory literature. In my view, it is. The key effect reported here is not an effect of choice alone, but shows that the value of unchosen options can be modified by outcomes associated with chosen options—even when those outcomes are delayed, in time, relative to the choice period. The effects are very robust with a pre-registered study and multiple replications—so the core result is overwhelmingly demonstrated. The paper is also very well written and enjoyable to read. Overall, I am very positive about the manuscript and only have a few minor comments/questions.

1. One question I have, which is likely hard to answer, is how much of the bias for unchosen options is constructed during the final decision phase. Is this a deliberative process where subjects consciously recall the pairs and make an inference? The fact that there was a 2.5s time limit for responses helps address this concern. Though, it might be worth reporting RTs (if they were recorded?) for each pair type and looking at whether the tendency to show inverse inference was related to (longer) RTs?

2. I was at first quite surprised at how robust some of the effects were (even for the S-chosen items) but then when getting to the Methods, it is explained that the number of pairs is relatively small (12), there are multiple deliberations (3) per pair, and participants are asked to explicitly write a description of why they chose one stimulus over another. So, the deliberation is pretty extensive and really drills the pairs into memory. Likewise, the outcome learning was a bit heavy-handed in that there were 6 repetitions and subjects had to press a key each time to indicate the outcome. While I do not think this heavy-

handed procedure undermines any of the claims, I think it would be useful to acknowledge these aspects of the design earlier in the paper (before the reader gets to the Methods) as one of my main reactions as I was reading it was that I was shocked that a single decision followed by a single outcome could have much influence even for the chosen items. The fact that the pairs and outcomes were overlearned also may have increased the tendency for subjects to use explicit strategies during the critical phase when choosing between the two unchosen options.

Reviewer #2 (Remarks to the Author):

Review for “How we learn to value the road not taken”, by Biderman and Shohamy. The authors investigate (over five experiments featuring > 600 participants) whether the act of deliberation affect the valuation of the unchosen option and found this being the case in a sort of “counterfactual normalization” process where, if the chosen option has been successful, the unchosen option is downvaluated. In their terms, the chosen and the unchosen option values are “inversely related”. The result is true on average, but the relation is also found at the inter-individual level ($P(\text{select } S+) / \text{Inverse relation}$). The effect is also linked to memory accuracy. The replicate the effect in several experiments, while concomitantly assessing the effect of number of repetitions, reward magnitude and painter identity. The paper addresses an important and timely question with a clever design and an outstanding methodological rigor. I am certain this paper will generate a lot of follow-up studies. Myself, I can't wait to check whether it replicates in my own data.

The editors of this journal know me for generally being a harsh reviewer, but I am afraid this time I am going to disappoint them. In this case I really have nothing to say to further improve this paper. Even the figures are beautiful. I can only plaud the authors for such an elegant and methodologically exemplar work.

Reviewer #3 (Remarks to the Author):

This is a very interesting paper, with a very elegant paradigm and a set of convincing results. Although I wish I had seen a bit more discussion on the relationship between episodic memory and counterfactual thinking (e.g., Schacter, Benoit et al, 2015, *Neurobiol Learn Mem*; Parikh and De Brigard, *Current Dir Psych Science*), as well as a bit more discussion on importantly similar paradigms in the choice-support bias literature (e.g., work by Henkel and Mather, which does get cited but, perhaps, not sufficiently discussed), the truth is that I did not find any reasons to suggest any major revisions of this manuscript. I think it offers convincing evidence for a mnemonic effect of an inverse relationship between the chosen and unchosen options that goes above and beyond the findings of the choice support bias. This paper makes an important contribution to both the memory and the decision-making literature. My comments, as such, are only minor.

p. 4. Ln. 120. Do we know whether participants remember, during the outcome learning stage, which choice they picked during the deliberation? There is some evidence to the effect that people may not remember which choice they picked after doing so. I take it that, because the number of options was small, and they were shown several times, that this isn't a concern. But thought I ask, anyways (I get back to this issue, below, when I comment on the methods).

p. 10. Ln. 321. I applaud this strategy to remove the 22 participants who used this inverse heuristic in the decision. I just mention this because, when I was reading the methods, this was the only "major" concern I thought of (I mention a "smaller" concern below, regarding retroactive inference, in reference to the methods). The authors anticipated it and dealt with it appropriately. Very nice.

p. 14. Last paragraph of Discussion. Here's another thing that the authors anticipated: what would happen if in the outcome learning the participant learns that they lost money, instead of no gain (i.e., if there is a S -, in addition to S+). The authors are correct in that upward counterfactuals are very typical, so one wonders what would happen when the association is with a forgone option that would have fare better. This, of course, is an open question for future research.

Another question that I think is worth asking—perhaps even a limitation of this study too—is what would happen with non-binary options. Deliberation often occurs between two choices, but it often involves more. Moreover, sometimes deliberation involves selecting the choices to consider. Work by Jonathan Phillips and Fiery Cushman, for instance, have suggested ways in which choices get selected from among a number of potential possible options. Their proposals are very congenial to the RL model advocated in this manuscript, and I think it could offer more ideas for potential follow-ups.

Minor comments about the methods:

p. 15, ln. 483-484. May be worth mentioning what the exclusion criteria were here, so that the reader, who already jumped in the text to the methods, does not have to further jump to the OSF site and then to the pre-registration to answer this question. Just a sentence or two would suffice.

p. 16. Ln. 522-525. The authors say: "We assumed that after making a choice, participants could still maintain both choice options in their working memory and feedback for the chosen option could transfer to the unchosen option just by virtue of being activated simultaneously in working memory." I share the authors' intuition that the participants may have been able to keep both choices in WM, particularly given the strong encoding conditions. But do you think that there may have been concerns with retroactive (and, maybe, proactive) interference here, so that unchosen options of previous pairs get wrongly associated with chosen (or unchosen) options of subsequent pairs and/or in subsequent deliberation blocks? Likely not, but just a thought.

Once again, what a nice paper. Thank you for inviting me to review it.

I am happy to be contacted should the authors have any questions about my comments. I can be reached at felipe.debrigard@duke.edu

Signed

Felipe De Brigard

Reviewer #4 (Remarks to the Author):

Biderman and Shohamy present a manuscript detailing how choice influences the value of unchosen options, specifically through a mechanism guided by episodic memory related processes. The authors show that there is a tendency to infer an opposing value to unchosen options based on the value associated with the chosen option. Further, this effect was associated with better associative memory performance both within and across participants. This study is very interesting, and opens a novel question about value formation for unchosen options. The study is strengthened by a clever experimental design, coupled with multiple replication data sets. The findings should have a large impact on individuals interested in interactions of memory and decision-making, counter-factual thinking, and choice-induced preference. Notably, I have previously reviewed this manuscript in a prior submission, and the authors have addressed some of my concerns. However, there are a few addressable concerns that remain in the manuscript, which I highlight below.

1. Regarding the experimental design, I was still confused as to the motivation for separating the feedback til the end of the choice session. Could the authors provide greater justification of this choice. I am partially concerned that this design may have “artificially” induced more episodic memory processes, given that this type of design would instill retrospective evaluation in memory as well as instructed learning (as opposed to feedback learning). In addition to providing additional justification, it would be helpful to provide further discussion of this design feature.

2. The relationship between explicit knowledge/memory and the choice behavior needs further discussion, especially given prior research from this lab group. Namely, in paradigms using sensory pre-conditioning and associative inference the authors have suggested that implicit memory has guided these processes, but here this process relies on more explicit memory processes. I think this facet of the findings need to be further discussed, especially given the parallels drawn to their prior work on acquired equivalence. What makes this decision-making context different? In what cases would individuals rely on explicit versus implicit associative memory processes. Relatedly, do the authors believe that the memory related binding that is occurring is happening during encoding or rather via inference during retrieval.

3. I think there are a few missed opportunities to connect these findings to a broader, highly relevant literature. Namely, the authors should discuss prior work characterizing episodic memory/decision

making interactions (Duncan/Shohamy, FeldmanHall/Murty, Bornstein/Daw, Palambo), the role of episodic memory in choice-induced preference (Salti et al, 2014), and the relationships between choice and subsequent episodic memory performance (Murty et al., 2015, DuBrow et al., 2019; Murty et al., 2019). I believe the acknowledgement of this prior work would greatly strengthen the current findings.

Thank you for the opportunity to revise our manuscript titled “How we learn to value the road not taken: Deliberation and memory shape the valuation of unchosen options”. We were very grateful for the reviewers’ thoughtful suggestions for improving the manuscript. Below we provide our responses, detailed point-by-point (in blue font). In the manuscript, revised text is marked with blue font.

Reviewer #1

This manuscript describes a series of behavioral studies in which subjects deliberated between pairs of artwork and then selected the piece to “send to auction.” In a subsequent phase, those selected pieces—but not the unselected pieces—were associated with outcome (either gaining money at auction, or not gaining money). In the next phase, subjects were then shown pairs of previously chosen or previously unchosen pictures and made a decision of which piece to add “to their portfolio.” For the previously-chosen pairs, one pair had gained money at auction (+) and one had not (0), so there was a straightforward choice—and subjects readily chose the pieces that had won money. For the unchosen pairs, one piece had previously been paired with a piece that had gained money (+) and the other piece had previously been paired with a piece that did not gain money (0). Here, there is a more interesting question of which piece subjects would choose. Across several studies, there was a clear and consistent tendency for subjects to systematically choose the piece that had previously been associated with a piece that did NOT gain money (and this was related to subjects’ memory for which pieces had originally occurred together). In some ways this is surprising as one might predict that a positive outcome for a chosen piece would transfer to the unchosen piece. But, the result is exactly the opposite (what the authors call inverse inference). The finding is highly reminiscent of—and consistent with—the literature on choice supportive memory (where there is a bias to remember chosen options more favorably and unchosen options less favorably). In fact, the biggest concern with the current manuscript is whether it is a sufficient advance beyond the choice supportive memory literature. In my view, it is. The key effect reported here is not an effect of choice alone, but shows that the value of unchosen options can be modified by outcomes associated with chosen options—even when those outcomes are delayed, in time, relative to the choice period. The effects are very robust with a pre-registered study and multiple replications—so the core result is overwhelmingly demonstrated. The paper is also very well written and enjoyable to read. Overall, I am very positive about the manuscript and only have a few minor comments/questions.

Thank you for this excellent summary and for your encouraging and supportive remarks. We found your comments to be very helpful and addressed them point by point below.

1. One question I have, which is likely hard to answer, is how much of the bias for unchosen options is constructed during the final decision phase. Is this a deliberative process where subjects consciously recall the pairs and make an inference? The fact that there was a 2.5s time limit for responses helps address this concern. Though, it might be worth reporting RTs (if they were recorded?) for each pair type and looking at whether the tendency to show inverse inference was related to (longer) RTs?

Thank you for this suggestion. We agree that understanding when value for unchosen options is constructed is a fascinating question. While the current study can not give a definitive answer to

this question, we added a new analysis of reaction time (RT) that speaks to it (below and p. 6), as well as discussion of this question (p. 15).

As suggested by the reviewer, we conducted new analyses to compare RTs for the different pair types and their relation to inverse inference. At the group-level, RTs for chosen pairs were faster than for unchosen pairs (see Figure 1a below), but there was no correlation between inverse inference and RT.

We did, however, find another interesting result inspired by the reviewer’s suggestion to look at RTs in this context. It is well known that choices for reward are faster than choices to avoid loss. Accordingly, we generally expect participants to be faster when they choose the more valuable painting and we can apply this logic to look at choices *within* the pairs of unchosen options: if participants did not update the value of unchosen pairs, RTs should be similar regardless of participants’ choices, but if they did update the value of unchosen options according to inverse inference, then we would expect *faster* RTs for unchosen options which were previously paired with a *no-reward*. Indeed, we found that when selecting among unchosen pairs, participants tended to be faster when selecting unchosen items previously paired with unrewarded items, compared to those previously paired with rewarded items (see Figure 1b below). RTs did not significantly modulate choices among chosen pairs, possibly because the tendency to select gain items was very high across participants so there was not enough variability in the dependent measure.

The pattern of RT in unchosen pairs mirrors the inverse decision bias in choice behavior and provides additional evidence for the tendency to infer the value of unchosen options in opposition to the chosen ones. The faster RTs to the no-gain choices in the Final Decisions phase also suggest that the positive value of these items was most likely already constructed before the subjects faced this choice (if the value was constructed on the spot during the decision phase, we would have expected longer RTs). We now include a brief description of these results in main text (p.6 and 15) and the full analysis in Supplementary Text 1.

Supplementary Figure 1. Reaction times in Final Decisions phase. (a) Participants were faster to decide among chosen pairs compared to unchosen pairs. (b) Choices in the Final Decisions phase were modulated by reaction times in unchosen pairs, but not in chosen pairs. In unchosen pairs, participants were faster when selecting items previously paired with unrewarded items, as indicated by the substantial positive slope in a multilevel Bayesian logistic regression predicting the probability to select rewarded items as a function of pair type and normalized reaction times (using z-score). In panel **a**, error bars denote standard error of the mean and points denote individual means. In panel **b**, the coefficients and model fits denote median estimation and 95% highest density interval of the posterior distribution.

2. I was at first quite surprised at how robust some of the effects were (even for the S-chosen items) but then when getting to the Methods, it is explained that the number of pairs is relatively small (12), there are multiple deliberations (3) per pair, and participants are asked to explicitly write a description of why they chose one stimulus over another. So, the deliberation is pretty extensive and really drills the pairs into memory. Likewise, the outcome learning was a bit heavy-handed in that there were 6 repetitions and subjects had to press a key each time to indicate the outcome. While I do not think this heavy-handed procedure undermines any of the claims, I think it would be useful to acknowledge these aspects of the design earlier in the paper (before the reader gets to the Methods) as one of my main reactions as I was reading it was that I was shocked that a single decision followed by a single outcome could have much influence even for the chosen items. The fact that the pairs and outcomes were overlearned also may have increased the tendency for subjects to use explicit strategies during the critical phase when choosing between the two unchosen options.

Thank you for emphasizing this point, which dovetails with a concern raised by Reviewer #4 about how delayed feedback might have artificially led to use of episodic processes (see below). We added a short description of our methodological rationale in the introduction of the paradigm (p. 4) and we also address this point in the revised Discussion (p. 14-15).

We designed the study to assess the effects of deliberation and memory on the valuation of unchosen options. It was therefore important for us to verify that participants actually deliberated on these decisions and learned the deliberation pairings and the outcomes of the chosen items. Following prior work in our lab (e.g. Wimmer and Shohamy, 2012), we repeated the pairings and outcomes to assure they were learned sufficiently, so that we would be in a strong position to assess inference.

Having observed and replicated a bias to estimate unchosen options in opposition to chosen ones, with sufficient deliberation and memory, we agree that an interesting next step will be to determine some of the boundary conditions of this effect. One interesting question is how this phenomenon will be affected if the deliberation is not as extensive or if the outcomes are provided immediately after the decision was made, rather than later on. Notably, in Experiment 2, we manipulated the memory strength of the pairs by changing the number of times they were repeated (two vs. six times). Pairs that were repeated more often were remembered better than those repeated less often. Critically, however, there was no difference in decision bias for

unchosen pairs between these conditions, nor was there any difference in the correlation of decision bias with memory.

Reviewer #2

Review for “How we learn to value the road not taken”, by Biderman and Shohamy. The authors investigate (over five experiments featuring > 600 participants) whether the act of deliberation affect the valuation of the unchosen option and found this being the case in a sort of “counterfactual normalization” process where, if the chosen option has been successful, the unchosen option is downvaluated. In their terms, the chosen and the unchosen option values are “inversely related”. The result is true on average, but the relation is also found at the inter-individual level ($P(\text{select } S+) / \text{Inverse relation}$). The effect is also linked to memory accuracy. The replicate the effect in several experiments, while concomitantly assessing the effect of number of repetitions, reward magnitude and painter identity. The paper addresses an important and timely question with a clever design and an outstanding methodological rigor. I am certain this paper will generate a lot of follow-up studies. Myself, I can’t wait to check whether it replicates in my own data.

The editors of this journal know me for generally being a harsh reviewer, but I am afraid this time I am going to disappoint them. In this case I really have nothing to say to further improve this paper. Even the figures are beautiful. I can only plaud the authors for such an elegant and methodologically exemplar work.

Thank you so much! We are honored and deeply appreciative of these supportive comments!

Reviewer #3

This is a very interesting paper, with a very elegant paradigm and a set of convincing results. Although I wish I had seen a bit more discussion on the relationship between episodic memory and counterfactual thinking (e.g., Schacter, Benoit et al, 2015, *Neurobiol Learn Mem*; Parikh and De Brigard, *Current Dir Psych Science*), as well as a bit more discussion on importantly similar paradigms in the choice-support bias literature (e.g., work by Henkel and Mather, which does get cited but, perhaps, not sufficiently discussed), the truth is that I did not find any reasons to suggest any major revisions of this manuscript. I think it offers convincing evidence for a mnemonic effect of an inverse relationship between the chosen and unchosen options that goes above and beyond the findings of the choice support bias. This paper makes an important contribution to both the memory and the decision-making literature. My comments, as such, are only minor.

Thank you so much for these positive, supportive, and constructive comments and for pointing us towards these papers. We have expanded the background on choice-supportive bias literature in the Discussion (p. 13) and also included a new paragraph in the Discussion about counterfactual thinking and episodic memory (p. 15).

p. 4. Ln. 120. Do we know whether participants remember, during the outcome learning stage, which choice they picked during the deliberation? There is some evidence to the effect that

people may not remember which choice they picked after doing so. I take it that, because the number of options was small, and they were shown several times, that this isn't a concern. But thought I ask, anyways (I get back to this issue, below, when I comment on the methods).

The Outcome Learning phase included only the chosen items, alongside either gain or no-gain. Unfortunately, our design does not allow us to assess whether participants retrieved the entire deliberation context (including the item they did not choose) during Outcome Learning. We agree this is an interesting question and plan to test this in a future study, for example, by presenting a source retrieval test, and/or using category specific images and looking for evidence for neural reactivation (following work we have done before, in Wimmer and Shohamy, 2012).

That said, we did test choice memory during the Surprise Memory Test, in which we presented pairs of paintings and asked participants whether each pair was an intact pair from the Deliberation phase or a recombined pair (pairs that included a chosen painting and an unchosen painting which did not appear together during Deliberation). If participants responded "intact", we also asked them which of the two paintings they chose during Deliberation. With data from this memory test we can look at overall choice memory; that is, whether participants remembered their chosen items regardless of whether they remembered the pairings from Deliberation (computing choice accuracy across hit and false alarm trials). Indeed, we find that participants remember their choices very well ($M = 0.84 \pm 0.02$) and in Experiment 1 choice memory predicted inverse decision bias ($\beta = 0.21 [0.04, 0.39]$). We now include this analysis in the Supplement (see Supplementary Text 2 and Supplementary Tables 1 and 6).

p. 10. Ln. 321. I applaud this strategy to remove the 22 participants who used this inverse heuristic in the decision. I just mention this because, when I was reading the methods, this was the only "major" concern I thought of (I mention a "smaller" concern below, regarding retroactive inference, in reference to the methods). The authors anticipated it and dealt with it appropriately. Very nice.

Thank you! We now include a short description of our removal strategy in the Method section (p. 19).

p. 14. Last paragraph of Discussion. Here's another thing that the authors anticipated: what would happen if in the outcome learning the participant learns that they lost money, instead of no gain (i.e., if there is a S -, in addition to S+). The authors are correct in that upward counterfactuals are very typical, so one wonders what would happen when the association is with a forgone option that would have fare better. This, of course, is an open question for future research.

We look forward to exploring this in the future!

Another question that I think is worth asking—perhaps even a limitation of this study too—is what would happen with non-binary options. Deliberation often occurs between two choices, but it often involves more. Moreover, sometimes deliberation involves selecting the choices to consider. Work by Jonathan Phillips and Fiery Cushman, for instance, have suggested ways in which choices get selected from among a number of potential possible options. Their proposals

are very congenial to the RL model advocated in this manuscript, and I think it could offer more ideas for potential follow-ups.

We thank the reviewer for pointing us to this potential follow-up. We absolutely agree that exploring the inference of value in non-binary decisions could provide important insight to how people reason about the interplay between the value of chosen and unchosen options across different contexts. We now included a paragraph in the Discussion that puts forward this possibility (p. 16).

Minor comments about the methods:

p. 15, ln. 483-484. May be worth mentioning what the exclusion criteria were here, so that the reader, who already jumped in the text to the methods, does not have to further jump to the OSF site and then to the pre-registration to answer this question. Just a sentence or two would suffice.

Thank you. We moved the exclusion criteria from the Supplement to the main text, as suggested (p. 17).

p. 16. Ln. 522-525. The authors say: “We assumed that after making a choice, participants could still maintain both choice options in their working memory and feedback for the chosen option could transfer to the unchosen option just by virtue of being activated simultaneously in working memory.” I share the authors’ intuition that the participants may have been able to keep both choices in WM, particularly given the strong encoding conditions. But do you think that there may have been concerns with retroactive (and, maybe, proactive) interference here, so that unchosen options of previous pairs get wrongly associated with chosen (or unchosen) options of subsequent pairs and/or in subsequent deliberation blocks? Likely not, but just a thought.

Thank you for bringing this point up. Indeed, because we present multiple pairs, it is possible that there may have been retroactive/proactive interference such that the chosen and unchosen were wrongly associated. However, we still see that as memory for the pairs is increased, so is the evidence for decision bias. In the future it will be interesting to assess how much of the variance we observe in decision bias is related to forgetting and proactive/retroactive interference. We now include this point in the Discussion (p. 13).

Reviewer #4

Bideman and Shohamy present a manuscript detailing how choice influences the value of unchosen options, specifically through a mechanism guided by episodic memory related processes. The authors show that there is a tendency to infer an opposing value to unchosen options based on the value associated with the chosen option. Further, this effect was associated with better associative memory performance both within and across participants. This study is very interesting, and opens a novel question about value formation for unchosen options. The study is strengthened by a clever experimental design, coupled with multiple replication data sets. The findings should have a large impact on individuals interested in interactions of memory and decision-making, counter-factual thinking, and choice-induced preference. Notably, I have

previously reviewed this manuscript in a prior submission, and the authors have addressed some of my concerns. However, there are a few addressable concerns that remain in the manuscript, which I highlight below.

We are very grateful for your (dual) feedback and constructive comments. Below we describe how we addressed each of your points.

1. Regarding the experimental design, I was still confused as to the motivation for separating the feedback til the end of the choice session. Could the authors provide greater justification of this choice. I am partially concerned that this design may have “artificially” induced more episodic memory processes, given that this type of design would instill retrospective evaluation in memory as well as instructed learning (as opposed to feedback learning). In addition to providing additional justification, it would be helpful to provide further discussion of this design feature.

Thank you for providing us with the opportunity to elaborate on this point. We added a new paragraph in the Discussion that explains our methodological choices (p. 14-15).

Briefly, and as also mentioned in response to a related comment by Reviewer #1 (see above), our overall approach was to optimize the task design to be in a strong position to address the main question of how deliberation and memory affect the valuation of unchosen options. With this goal in mind, we felt it was critical to verify that deliberation and memory would be robust. Similarly, we kept the outcome learning in a separate phase for the same purpose: verifying that participants would use the long-term memory associations they formed during Deliberation to transfer value, rather than doing it while the two options are kept in working memory, as would be possible if the feedback is immediate. Now that our findings established that memory modulates the inverse inference of value of unchosen options, we are in a position to go back and probe this effect further and determine its boundary conditions, for example by asking what happens when the feedback is provided immediately, as in standard RL tasks.

2. The relationship between explicit knowledge/memory and the choice behavior needs further discussion, especially given prior research from this lab group. Namely, in paradigms using sensory pre-conditioning and associative inference the authors have suggested that implicit memory has guided these processes, but here this process relies on more explicit memory processes. I think this facet of the findings need to be further discussed, especially given the parallels drawn to their prior work on acquired equivalence. What makes this decision-making context different? In what cases would individuals rely on explicit versus implicit associative memory processes. Relatedly, do the authors believe that the memory related binding that is occurring is happening during encoding or rather via inference during retrieval.

Thank you for giving us the opportunity to add detail regarding this point. We agree that the question about implicit vs. explicit influences of memory on decision-making is an important albeit elusive one and have added a discussion of this issue (p. 13). Indeed, in prior studies, we often find that memory’s influence on value-based decisions appears to happen outside of conscious awareness and that explicit memory for associations is orthogonal to their influence on choice. Here, we find evidence, repeatedly, that explicit memory for the association between items is indeed related to choice.

We do not yet have a complete understanding of the factors that determine when this relationship is implicit vs. explicit, but we think the current findings point to some possibilities that can be explored in future work. First, the nature of the association in the current study is one that is formed in the context of deliberation towards a decision. That is, the association between the two choice options requires active and deep encoding if one is to reach a decision. This contrasts with sensory preconditioning or acquired equivalence, where the association is incidental and passive, and not integration to any active decision or action the subject must take.

Second, related to the reviewer's earlier point (#1), in this study we sought to verify that memory and deliberation would be robust enough to be able to investigate their role in value transfer. As noted, it is possible that by reaching that threshold of memory and deliberation we also increase the likelihood that memory would be explicit. Although, it is worth pointing out that the interesting finding that replicated across our versions was not just that there was robust memory, but, critically, that memory was correlated with the inverse inference effect.

In future work we hope to further explore this question by (a) manipulating instructions that emphasize vs de-emphasize awareness of the relationship between memory and decisions; (b) varying the temporal gap between deliberation and decisions and testing how weakening of the initial memory may change the relationship between memory and inverse bias.

Finally, our hypothesized mechanism focusses on encoding processes. This is difficult to test in behavior alone, although the new RT findings (described in response to R1 point 1, see above) hint to the possibility of value transfer during encoding. Choices were faster for no-gain unchosen options, pointing to the possibility that these choice options already gained their positive value prior to the decision phase. These results appear briefly in the Results section (p. 6) and in the Supplement (Supplementary Text 1) and are discussed on p. 15.

3. I think there are a few missed opportunities to connect these findings to a broader, highly relevant literature. Namely, the authors should discuss prior work characterizing episodic memory/decision making interactions (Duncan/Shohamy, FeldmanHall/Murty, Bornstein/Daw, Palambo), the role of episodic memory in choice-induced preference (Salti et al, 2014), and the relationships between choice and subsequent episodic memory performance (Murty et al., 2015, DuBrow et al., 2019; Murty et al., 2019). I believe the acknowledgement of this prior work would greatly strengthen the current findings.

Thank you for inspiring us to expand our discussion. We now include a new paragraph in the Discussion that explores the previous studies about the interaction between memory and decision making, specifically, about the parameters that push people to consult specific past events when evaluating their next choices, and the effects of agency on subsequent memory of the decisions (p. 14). Additionally, we refer to the relationship between episodic memory and choice-induced preference change in the paragraph that discusses the similarities and differences between our study and choice-induced preference change studies (p. 14).

REVIEWERS' COMMENTS

Reviewer #1 (Remarks to the Author):

My initial evaluation of the manuscript was very positive (as were the evaluations from most of the other Reviewers). I only had a couple of minor suggestions/questions and I am fully satisfied with the authors' responses. In fact, they have added a nice, supporting analysis looking at reaction times when deciding between unchosen pairs. Again, this is a very interesting, rigorous, and well written manuscript and will be of broad interest to readers. I have no further suggestions or concerns.

Reviewer #2 (Remarks to the Author):

I made no comment during the first round, this paper is very good.

Reviewer #3 (Remarks to the Author):

This is an excellent manuscript. I have no further concerns.

Reviewer #4 (Remarks to the Author):

The authors did a commendable job addressing my concerns, and I was quite enthusiastic about the paper upon my first read and even more enthusiastic after these revisions! The extended discussion really bridged these findings both with the authors prior work and the field more generally. Further, the RT analysis proposed by reviewer 1 is really interesting. Great work!

We are thrilled about the positive decision regarding our manuscript. We are very grateful for the reviewers' great suggestions to improve the manuscript throughout the reviewing process and we are glad they found our revision to be adequate for publication.

Reviewer #1

My initial evaluation of the manuscript was very positive (as were the evaluations from most of the other Reviewers). I only had a couple of minor suggestions/questions and I am fully satisfied with the authors' responses. In fact, they have added a nice, supporting analysis looking at reaction times when deciding between unchosen pairs. Again, this is a very interesting, rigorous, and well written manuscript and will be of broad interest to readers. I have no further suggestions or concerns.

We thank you for the great suggestions to improve the manuscript, and for directing us towards a useful reaction times analysis.

Reviewer #2

I made no comment during the first round, this paper is very good.

Thank you!

Reviewer #3

This is an excellent manuscript. I have no further concerns.

Thank you! We appreciate the positive feedback and the useful suggestions to improve the manuscript.

Reviewer #4

The authors did a commendable job addressing my concerns, and I was quite enthusiastic about the paper upon my first read and even more enthusiastic after these revisions! The extended discussion really bridged these findings both with the authors prior work and the field more generally. Further, the RT analysis proposed by reviewer 1 is really interesting. Great work!

We are thrilled that we were able to address your concerns and appreciate your feedback throughout the reviewing process.